# Anti-cancer compound screening identifies Aurora Kinase A inhibition as a means to favor CRISPR/Cas9 gene correction over knock-out

Danny Wilbie[1], Selma Eising[2], Vicky Amo-Addae[2], Johanna Walther[1], Esmeralda Bosman[1], Zhiyong Lei[3], Olivier G de Jong[1], Jan J Molenaar[1,2], Enrico Mastrobattista[1]*

**1** Department of Pharmaceutics, Utrecht Institute for Pharmaceutical Sciences (UIPS), Utrecht University, Utrecht, the Netherlands, **2** Princess Máxima Center for Pediatric Oncology, Utrecht, the Netherlands, **3** CDL Research, University Medical Center Utrecht, Utrecht, the Netherlands

* e.mastrobattista@uu.nl (EM)

## Abstract

CRISPR gene therapy holds the potential to cure a variety of genetic diseases by causing a targeted DNA break, which is repaired by host DNA damage responses. One option to introduce precise gene corrections is via the homology-directed repair (HDR) pathway. The problem in utilizing this pathway is that CRISPR-induced double stranded DNA breaks are more likely to be erroneously repaired by the non-homologous end joining (NHEJ) pathway, which may introduce random insertions or deletions at the cut site. We screened a small library of oncological drug compounds to steer the DNA repair process towards preferential HDR activation. We included forty compounds in the screen based on their mechanism of action. After optimizing the toxicity and adding these compounds during gene editing, nine showed a potential benefit for HDR activation. Three were shown to be beneficial after validation: rucaparib, belinostat and alisertib. The Aurora Kinase A inhibitor alisertib in particular led to an over 4-fold increase in preferential gene correction over gene knock-out in two cell models (HEK293T and Hepa 1–6) at sub-micromolar dosages on the eGFP locus, prompting further validation. On the long term this pathway did show cytotoxicity especially in the HEK293T cells, indicating further mechanistic investigation is needed, but this toxicity was less pronounced in primary hepatocytes.

## Introduction

Curative gene editing by RNA-guided CRISPR/Cas9 nucleases has progressed into clinical trials in recent years, with applications varying from *ex vivo* cell modification to *in vivo* gene editing [1,2]. This gene therapy method utilizes the CRISPR-associated protein 9 (Cas9) endonuclease, an enzyme which complexes with a guide RNA sequence which can direct it to a specific DNA target. This ribonucleoprotein (RNP)

**Data availability statement:** All unprocessed data related to this manuscript with the relevant metadata can be found at: https://doi.org/10.24416/UU01-AYSRSY.

**Funding:** This work was funded by the Netherlands Organisation for Scientific Research (NWO) Talent program VICI, grant number 865.17.005, which was granted to E.M. Url: https://www.nwo.nl/. The funder did not play any role in the study design, data collection and analysis, decision to publish, or preparation of the manuscript.

**Competing interests:** The authors have declared that no competing interests exist.

complex binds to the DNA sequence complementary to the guide RNA, after which the Cas9 nuclease causes a double stranded DNA break (DSB). In the context of gene editing in cells, the RNP needs to reach the cell nucleus and bind to its target in the genomic DNA. Cells are equipped with DNA repair pathways to resolve this damage, which can be exploited for therapy [3].

DSBs are predominately repaired by the non-homologous end-joining (NHEJ) and homology directed repair (HDR) pathways [4]. NHEJ leads to ligation of the broken DNA ends by DNA ligase 4, which is often repaired perfectly but can also introduce small insertions or deletions (indels) at the double stranded break site. When repaired faithfully, this ligation leads to recovery of the Cas9 target sequence, which results in a cycle of DNA cutting and repair while the RNP complex remains present. NHEJ is therefore eventually error prone, and may lead to indels which can shift the reading frame of the protein encoded by that gene [5,6]. This process functionally knocks out the encoded protein, which is therapeutically beneficial when a protein is overexpressed or has mutations through which the protein gained a pathogenic function. Therapeutically employing this mechanism is, for example, under clinical evaluation for the treatment of transthyretin amyloidosis through intravenous injection of LNP-formulated Cas9 mRNA and sgRNA [7]. HDR, in contrast, causes partial resection of the broken DNA strand and uses a homologous DNA strand as template to guide the repair. This process naturally uses the sister chromatid during mitosis as the repair template. The exact mechanisms have been summarized excellently in other works [8–10]. HDR can be exploited using an artificial DNA template to introduce specific mutations into a gene, and can therefore be used to repair damaged genes in genetic disorders. HDR has been used in this way to resolve point mutations as well as inserting larger DNA sequences [11–13].

However, HDR has proven to be difficult to translate to an effective gene therapy. HDR occurs less frequently than NHEJ, due to the relatively low expression of the effector proteins for HDR compared to NHEJ (8). Notably, HDR is active in the late S, G2 and early M phases of mitosis, but practically absent during other cell cycle phases [6,14,15]. Furthermore NHEJ is always active and out-competes the HDR machinery even during mitosis, leading to the odds of faithful gene correction to be low. This NHEJ preference by cells hampers clinical translatability of HDR, as the majority of treated cells will undergo the incorrect repair pathway and exhibit unwanted indels at the target site. Those cells are then no longer easy to target by CRISPR-Cas, as the target DNA sequence has mutated in an unpredictable manner and is now heterogeneous between edited cells. While autologous gene-corrected cells have recently entered clinical trials, this drawback has led the field to consider alternative gene-editing tools for direct *in vivo* injection of gene correction machinery such as base editors [16–18].

Prominent novel developments towards this include the Base-Editor and Prime-Editor systems. These cause single stranded DNA breaks and contain an additional effector protein domain fused to the Cas9 scaffold. Base editors chemically modify nucleic acids through their enzymes, whereas prime editor uses a reverse transcriptase and a modified gRNA molecule to write mutations into the genome

directly [19]. The range of mutations these systems can resolve is in theory limited compared to HDR as these systems cannot facilitate large insertions, but preliminary results show that the specificity of the gene correction is much higher, especially for point mutations [20]. These developments pose the question whether HDR mediated gene correction for small mutations is relevant for clinical development, possibly with add-on therapies to enhance its specificity.

Due to the aforementioned competition between HDR and NHEJ, it is essential that the repair pathway is shifted towards preferential or exclusive HDR before it can be safely used clinically. Many groups have demonstrated that small molecule compounds influencing the DNA repair pathways or the cell cycle are capable of improving HDR as recently reviewed by Shams et al. [10]. Primarily efforts were done specifically on utilizing both NHEJ inhibitors and HDR enhancers. Prominent examples include SCR-7, a DNA ligase 4 inhibitor, which inhibits NHEJ and has been demonstrated to result in HDR becoming the dominant pathway both *in vitro* and *in vivo* [10,21,22]. Direct HDR enhancement can be achieved by RS-1, which stabilizes the active conformation of Rad51, which is a limiting factor in HDR progression. This compound shows similar success as SCR-7 [10,23]. Furthermore, alternative strategies utilizing HDR such as *in* trans Cas9 nickases can be utilized to improve the outcome of gene editing [24]. While the *in vivo* data is promising, neither compound is in clinical development, making information on use in humans sparse.

The rationale of this work was therefore to explore a selection of clinically-tested drugs for potential CRISPR-modulating properties, to aid clinical development in future applications of HDR. We focused on drugs that are able to target DNA repair pathway regulation, signaling for cell proliferation and genomic instability in general [25]. Interestingly, many therapies that are designed for cancer modulate proteins in these domains, as these proteins have an important role in both cancer and mechanisms involved in CRISPR gene editing. Therefore the aim was to screen oncological drugs to find novel modulators and pathways to enhance CRISPR HDR and enable potential add-on therapies in the future, and to add on to the growing toolkit of CRISPR enhancers used in the laboratory setting with clinically relevant drug molecules.

## Materials and methods

### HEK293T-eGFP and Hepa 1–6-eGFP cell culture conditions

HEK293T cells with constitutive enhanced green fluorescent protein (eGFP) expression (HEK293T-eGFP [26]) were cultured as described previously using low glucose DMEM containing 10% fetal bovine serum (Sigma-Aldrich, Zwijndrecht, The Netherlands) [12]. Hepa 1–6-eGFP cells were cultured in high-glucose DMEM (Sigma-Aldrich) supplemented with 10% fetal bovine serum (Sigma-Aldrich). Cell culture plastics were acquired from Greiner Bio-One (Alphen aan de Rijn, The Netherlands).

Unless specified otherwise, gene editing experiments for both cell lines were conducted by seeding cells in 96-well Greiner CellStar plates (Greiner Bio-One) at a density of 3 x $10^5$ cells/cm$^2$. The same cell density was applied in other well plate formats. Medium was supplemented with 1x antibiotic/antimycotic solution (Sigma-Aldrich) during gene-editing experiments, 48 hours after adding genome editing formulations.

### Hepa 1–6 eGFP cell line construction

Hepa 1–6 cells were graciously donated by dr. Piter Bosma from the Tytgat Institute for Liver and Intestinal Research, Amsterdam University Medical Centers. These cells were stably transduced using a lentiviral vector to constitutively express eGFP. Lentiviral particles carrying the eGFP gene were generated by co-transfection of a functional eGFP gene in the pHAGE2-EF1a-IRES-PuroR lentiviral vector, alongside the pMD2.G plasmid and PSPAX2 plasmid (Addgene #12259 and #12260, respectively) at a 2:1:1 ratio in HEK293T cells using 3 µg polyethylenimine (25 kDa linear, Polysciences, Warrington, USA) per µg plasmid DNA. The supernatant of these cells was cleared of cells by five minutes of centrifugation at 500 x *g*, followed by 0.45 µm syringe filter filtration. Lentiviral supernatants were stored at −80 °C until further use. Transduction was performed overnight at a multiplicity of infection of 0.1. Puromycin selection was performed using 2 µg/mL puromycin (InvivoGen, San Diego, USA) to the culture medium 48 hours post transduction. After two weeks of

puromycin selection, eGFP expressing cells were sorted on a BD FACSAria III cell sorter (Becton Dickinson, New Jersey, USA), and subsequently expanded for 2 weeks prior to experimental use.

## Drug compound addition

A selection of forty small molecule drug compounds was made from the in-house oncological library of the high-throughput screening facility of the Princess Màxima Center. The selection was based on the mechanism of action of the drugs predicted to influence CRISPR gene-editing outcomes. An overview of the compounds used in this study is given in Table 1. These compounds, dissolved in DMSO at 10 mM, were added to wells using the Echo 550 liquid handler (Beckman Coulter, Woerden, The Netherlands) for the large compound screen and the TECAN D300e digital dispenser (Tecan Group LTD, Männedorf, Switzerland) for the subsequent validation experiments. Cells were seeded in sterile cell culture plates pre-primed with the compounds calculated to yield the correct concentrations in each well. The concentration of DMSO was normalized in each well to 0.1% for all experiments and conditions in this work unless specified otherwise.

## Cytotoxicity assays

Forty microliters of a HEK293T-EGFP cell suspension (3000 cells/well) were plated in tissue-culture treated flat-bottom 384-well microplates (catalogue number 3764, Corning, New York, USA) using a Multidrop Combi Reagent Dispenser

**Table 1. Selected compounds and their respective IC50 in HEK293T-eGFP cells after 3 days of incubation. The dosage schemes in the gene-editing screening experiment were based on these IC50 values, as indicated by color. Compounds marked green had no measurable toxicity below 10 µM.**

**Legend and dosing scheme (µM)**

| High dose | Medium dose | Low dose | Compound | Target | IC50 (µM) |
|---|---|---|---|---|---|
| 0.001 | 0.0001 | 0.00001 | Abemaciclib | CDK4 & −6 | 1,07 |
| 0.01 | 0.001 | 0.0001 | Belinostat | HDAC 1–11 | 1,10 |
| 0.1 | 0.01 | 0.001 | GSK1070916 | AURKB & -C | 1,39 |
| 1 | 0.1 | 0.01 | Molibresib | BRD4 | 1,78 |
| 10 | 1 | 0.1 | LMK-235 | HDAC4 & −5 | 1,79 |
|  |  |  | Ceralasertib | ATR | 1,84 |
| **Compound** | **Target** | **IC50 (µM)** | Vorinostat | HDAC 1–11 | 3,34 |
| Paclitaxel | TUBB | 0.00312 | Entinostat | HDAC 1–11 | 3,56 |
| Prexasertib | CHEK1 | 0.00533 | Pevonedistat | NAE1 | 4,74 |
| GSK461364 | PLK1 | 0.00835 | I-BRD9 | BRD9 | 8,94 |
| Romidepsin | HDAC 1 & −2 | 0.0164 | Alisertib | AURKA | >10 |
| Volasertib | PLK1 | 0.0245 | CPI-455 | pan-KDM5 | >10 |
| Panobinostat | HDAC 1–11 | 0.0306 | Epidaza | HDAC 1–3 | >10 |
| THZ1 | CDK7 | 0.0371 | GSK2830371 | WIP1 | >10 |
|  |  |  | JQ-1 COOH | BRD4 | >10 |
| Adavosertib | WEE1 | 0.227 | KU-55933 | ATM | >10 |
| CYC065 | CDK2 & −3 | 0.255 | KU-60019 | ATM | >10 |
| Berzosertib | ATR | 0.335 | Olaparib | PARP1 & −2 | >10 |
| THZ531 | CDK12&13 | 0.352 | Pamiparib | PARP1 & −2 | >10 |
| AT7519 | CDK1 & −2 | 0.438 | PCI-34051 | HDAC 8 | >10 |
| Karonudib | MTH1 | 0.450 | Ribociclib | CDK4; CDK6 | >10 |
| Birabresib | BRD2–4 | 0.540 | Rucaparib | PARP1 & −2 | >10 |
| BI 894999 | BRD4 | 0.582 | TAK-580 | BRAF; RAF1 | >10 |
| CPI-203 | BRD4 | 0.651 | XAV-939 | TNKS1 & −2 | >10 |

(Thermo Scientific, Breda, The Netherlands). Cells were cultured for 16–24 hours under standard culturing conditions (5% $CO_2$, 37 °C). Subsequently, 100 nL of the drugs (in DMSO, at different concentrations) are added to the wells containing the cells, to yield final concentrations of 0.1 nM, 1 nM, 10 nM, 100 nM, 1 µM and 10 µM (0.25% DMSO). All dose ranges were added in duplicate, followed by 72 hours of incubation. Cell viability was determined using a tetrazolium based metabolic activity assay [27]. Briefly, 5 µL of 3-(4.5-dimethylthiazol-2- yl)-2,5-diphenyltetrazolium bromide (MTT) solution (5 mg/mL MTT in sterile PBS) was added per well, and the microplates were incubated for 4 hours at 37°C and 5% $CO_2$ in a cell culture incubator. Next, 40 µL of 10% SDS/0.01 M HCl was added per well, and the microplates were incubated for 24–72 hours at 37°C and 5% $CO_2$ in a cell culture incubator. Subsequently the absorbance at 570 nm and background absorbance at 720 nm were measured using the Spectramax i3x (Molecular Devices, San Jose, USA). Subsequently, the absorbance values at 720 nm were subtracted from the absorbance values at 570 nm, and the corresponding values were used to plot dose-response curves.

The data was normalized to the DMSO-treated cells (defined as 100% viability) and the empty controls (defined as 0% viability). IC50 values at 72 hours were calculated by determining the concentrations of the drug needed to achieve a 50% reduction in cell viability using the extension package *drc* in the statistic environment of R Studio (version 4.0.2) [28].

A narrower cytotoxicity range was determined by exposing cells to 0.1–1 µM of the tested compounds. 48 hours after treatment started, cells were washed and harvested using medium and one third of the volume was transferred to a fresh 96 well plate, analogous to how cells are treated during gene editing experiments. Subsequently, cells were cultured for another 3–5 days. In the case of 5 days, one third of the cells was again transferred into a fresh 96-well plate on day 3 to allow enough space for logarithmic cell growth during the additional incubation time. Cell viability was determined by the Promega One Step MTS assay (Promega, Madison, USA) using the manufacturer's specifications. Relative cell viability for the Hepa 1–6-eGFP cells was calculated by normalizing the compound conditions to controls treated with DMSO only or DMSO plus gene editing formulations. For the HEK293T cells, the absolute absorbance at 590 nm was used as it was more representative of the relative cell viability between samples.

Cell morphology was assessed using the Nikon Eclipse Ti2 microscope (Nikon Europe, Amstelveen, The Netherlands). Pictures were acquired at 10x magnification with a Nikon DSLR 10 camera using the same imaging settings within each experiment set (Nikon Europe, Amstelveen, The Netherlands).

Finally, cytotoxicity of alisertib was determined on primary murine hepatocytes using an MTT assay. Primary murine hepatocytes were harvested from Ai9 mice and cultured in 2D prior to transfection as previously reported [29]. Alisertib was removed after 48 hours of incubation. Toxicity was measured at this timepoint, and 72 hours later,. Cell viability was calculated by subtracting a medium blank from the absorption, and dividing the absorption by that of a vehicle-only control.

## CRISPR-Cas9 nanocarrier formulation

SpCas9 protein was produced and purified in-house as described previously [12]. sgRNA and HDR template DNA were acquired from Sigma-Aldrich (Haverhill, United Kingdom).

Lipid nanoparticles (LNP) carrying SpCas9, sgRNA and HDR template DNA were formulated using the components and molecular ratios described previously [12]. 1,1′-((2-(4-(2-((2-(bis(2-hydroxydodecyl)amino)ethyl)(2-hydroxydodecyl) amino)ethyl)piperazin-1-yl)ethyl)azanediyl)bis(dodecan-2-ol) (C12-200) was acquired from CordonPharma (Plank-stadt, Germany) and used as the ionizable lipid in the formulation. 1,2-dioleoyl-sn-glycero-3-phosphoethanolamine (DOPE) was acquired from Lipoid GmbH (Steinhausen, Switzerland), Cholesterol and 1,2-dimyristoyl-rac-glycero-3-methoxypolyethylene glycol-2000 (PEG-DMG) were acquired from Sigma-Aldrich, and 1,2-dioleoyl-3-trimethylammonium-propane (DOTAP) was acquired from Merck (Darmstadt, Germany). LNP were produced using microfluidic mixing with the Dolomite Microfluidics system (Dolomite Microfluidics, Royston, United Kingdom) and herring-bone micromixer chip with hydrophilic coating (Dolomite Microfluidics, catalogue number 3200401). A total flow rate of 1.5 mL/min and flow rate ratio

of 2:1 were used between an aqueous outer phase containing SpCas9, sgRNA and HDR template in nuclease free water, and the lipids in 100% ethanol in the inner phase, respectively. The resulting LNP were diluted 4 times in Dulbecco's PBS (Sigma-Aldrich). In the experiments using the Hepa 1–6 eGFP cells, ProDeliverIN CRISPR (OZ Biosciences, San Diego, USA) was used as reported previously [12].

## Compound screening to modulate CRISPR gene repair outcomes

Compounds were assessed in three dosages to assess the effects on gene-editing efficacy. The highest concentration was based on the IC50 of the compounds as determined by the cytotoxicity determination, with a medium and low dose which were 10 and 100 times diluted respectively compared to this highest dosage. Cells were incubated with these compounds for 24 hours prior to LNP addition. LNP were added to all wells at a final concentration of 20 nM SpCas9 to achieve robust genome editing [12]. After 24 hours of co-incubation, medium was refreshed and cells were transferred to a 48 well plate for further culturing. Six days after adding compounds, cells were processed for flow cytometric analysis of the gene knock-out and gene-correction efficiencies [12,30]. Briefly, a ssDNA template was used carrying two nucleotide mutations to convert the eGFP sequence to that of a blue fluorescent protein (BFP), as well as mutating the PAM sequence to ensure robust HDR. The sgRNA spacer sequence was 5'-GCUGAAGCACUGCACGCCGU-3', and the HDR template sequence was 5'-CAAGCTGCCCGTGCCCTGGCCCACCCTCGTGACCACCCTGAGCCACGGCGTGCAG TGCTTCAGCCGCTACCCCGACCACATGAAGC-3'.

Flow cytometry using the BD FACS Canto II (Becton Dickinson) was used to determine cells undergoing NHEJ (eGFP and BFP negative population) and HDR (eGFP negative, BFP positive). Data analysis was performed using Flowlogic (version 8.7). Graphpad PRISM (version 9.3.1) was used for statistical analysis and preparing graphs.

The percentage of HDR relative to total gene editing in a given cell population (hereafter named Relative HDR (% of edited cells) was calculated by dividing the absolute HDR population by the sum of gene-edited cells found in the BFP+ and eGFP- gates. Absolute HDR (% of all cells) and Absolute NHEJ (% of all cells) were analyzed where appropriate. The gating strategy is given in S1 Fig in S1 File.

## Gene sequencing and genotype analysis

For genotypic analysis, HEK293T-eGFP cells were pretreated with alisertib for 24 hours and concurrently treated with alisertib and CRISPR-Cas9 LNP for 48 hours prior to harvesting by trypsinization. 25% of the harvested cells were transferred to a fresh well plate for expansion and analysis by flow cytometry as described previously. The remaining cells were lysed and genomic DNA was extracted using the QIAamp DNA Blood Mini Kit (Qiagen, Venlo, The Netherlands) according to the manufacturer's instructions. PCR was performed to amplify the eGFP locus in the obtained DNA using the Phusion™ High-Fidelity DNA Polymerase (2 U/µL) (Thermo Fisher Scientific, Landsmeer, The Netherlands). The PCR mixture (50 µL) contained 200 ng of DNA, 0.5 µM of forward primer (5'- GACGTAAACGGCCACAAGTT – 3' (Integrated DNA Technologies Leuven, Belgium) and reverse primer (5'- CGATGTTGTGGCGGATCTTG – 3' (Integrated DNA Technologies Leuven, Belgium), 200 µM of dNTPs (dNTP Mix (10 mM each) (Thermo Fisher Scientific), 1 × Phusion HF buffer, 3% DMSO and 1 units of Phusion High Fidelity DNA polymerase. The DNA was amplified using the following thermocycling steps: 98°C for 30 sec; 35 cycles of 98°C for 10 sec, 62°C for 30 sec and 72°C for 30 sec; 72°C for 10 min. The PCR products were purified using the GeneJET PCR Purification Kit (Thermo Fisher Scientific). Sanger sequencing was performed by Macrogen (Amsterdam, The Netherlands) using the previously mentioned reverse primer as sequencing primer.

Sanger sequencing chromatograms were analyzed using the TIDER webtool (http://shinyapps.datacurators.nl/tider/) using default settings [31]. The reference chromatogram, corresponding to the blue fluorescent mutation, was generated from a gBlock gene fragment acquired from Integrated DNA technologies. The control (eGFP) chromatogram was generated from untreated cells. The indel frequencies up to −5 and +5 were plotted using Graphpad PRISM, version 9.3.1.

## Statistical analysis

Statistical analysis was conducted using Graphpad PRISM version 9.3.1 using the Two-way ANOVA method, comparing the intra-group effect where relevant in a multiple comparisons test with Šidák correction unless stated otherwise. IC50 values were calculated using the [antagonist] vs response function of Graphpad PRISM version 9.3.1 where applicable.

## Results

Compounds were selected from a clinically assessed oncological drug library used for drug discovery in pediatric cancer, as explained in Fig 1A. This library contained a large variety of drugs, prompting a selection to be made. The rational drug selection was done by defining groups of pathways expected to modulate CRISPR gene editing outcomes: cell cycle modulation, DNA damage repair modulation and chromatin modulation. Drugs which were not easily categorizable in a single domain were included in the screen as well, due to the potential of unexpected effects on the gene editing outcome. This selection led to 40 compounds to be screened, summarized in Table 1.

The cytotoxicity of the selected compounds was first assessed on the HEK293T-eGFP cell line using an MTT assay and determining the IC50 values of each compound after three days of treatment. These conditions were selected to mimic the maximum exposure time to the compound to be used in the screen (Table 1, S2 Fig in S1 File.). Three compounds showed considerable toxicity with IC50 values in the nanomolar range. The majority of compounds were tolerated in the micromolar range however, or were not toxic in the investigated concentration range. These toxicity values were used to dose the compounds in a sub-toxic dosage in subsequent gene-editing experiments, as noted in Table 1. The closest $^{10}$log concentration to the IC50 (high dose) and two $^{10}$log values below were used to preliminarily determine effects of the compounds on gene-editing efficiency, as the compound needed to be efficacious in a non-toxic or at most sub-toxic dose for potential therapeutic application. Cells were incubated with the compounds for 24 hours prior to adding lipid nanoparticles (LNP) carrying Cas9 enzyme, sgRNA and an HDR template designed to mutate the eGFP sequence to a blue fluorescent phenotype, as reported previously [12].

The effect of all screened compounds on gene editing are given in Fig 1B. Gene knockout (loss of eGFP signal) and correction (rise of BFP signal) were measured, as shown in S3 Fig in S1 File. From these values the relative HDR efficiency was calculated as the percentage of HDR in total gene edited cells, as shown in Fig 1B. The gating strategy used in the flow cytometry data analysis is given in S1 Fig in S1 File.

The effect of the compounds was compared to cells treated with only LNP containing RNP and HDR template DNA (n = 29 wells). A minimum of 1000 events in the single-cell gate was deemed necessary at a minimum for data analysis. Conditions not exceeding this number due to unexpected toxicity were excluded from Fig 1B. Compound treated cells deviating at least one standard deviation from the LNP-only control mean (dashed line in Fig 1B) were considered to differ relevantly from LNP alone, and were considered a potential hit for altering the gene-editing outcome selection. In this study, only compounds which increased the relative incidence of HDR were investigated further, other findings are summarized in S2 Fig. in S1 File. This was calculated by the amount of HDR events divided by all gene edited cells (blue fluorescent and non-fluorescent cells combined). Nine compounds showed at least one concentration above that threshold. Further conclusions were not taken from this screen as all datapoints were single measurements. Validation experiments were performed to confirm these hits in triplicate and in a narrower dosage range.

The hits were validated by narrowing the dose range between the most efficacious concentration found in the initial screen and the $^{10}$log-lower concentration in triplicate (Fig 2A). Three compounds showed a strong dose-dependent preferential activation of HDR over NHEJ compared to controls treated with LNP only (dotted line): rucaparib, belinostat and alisertib. The other compounds did not show a clear dose-dependent enhancement of HDR efficiency upon this further scrutiny. Further analysis was done on the two main gene editing outcomes of NHEJ and HDR. The three validated hits exhibited different effects on these two repair pathways as shown in Fig 2B. Rucaparib and alisertib both inhibited NHEJ

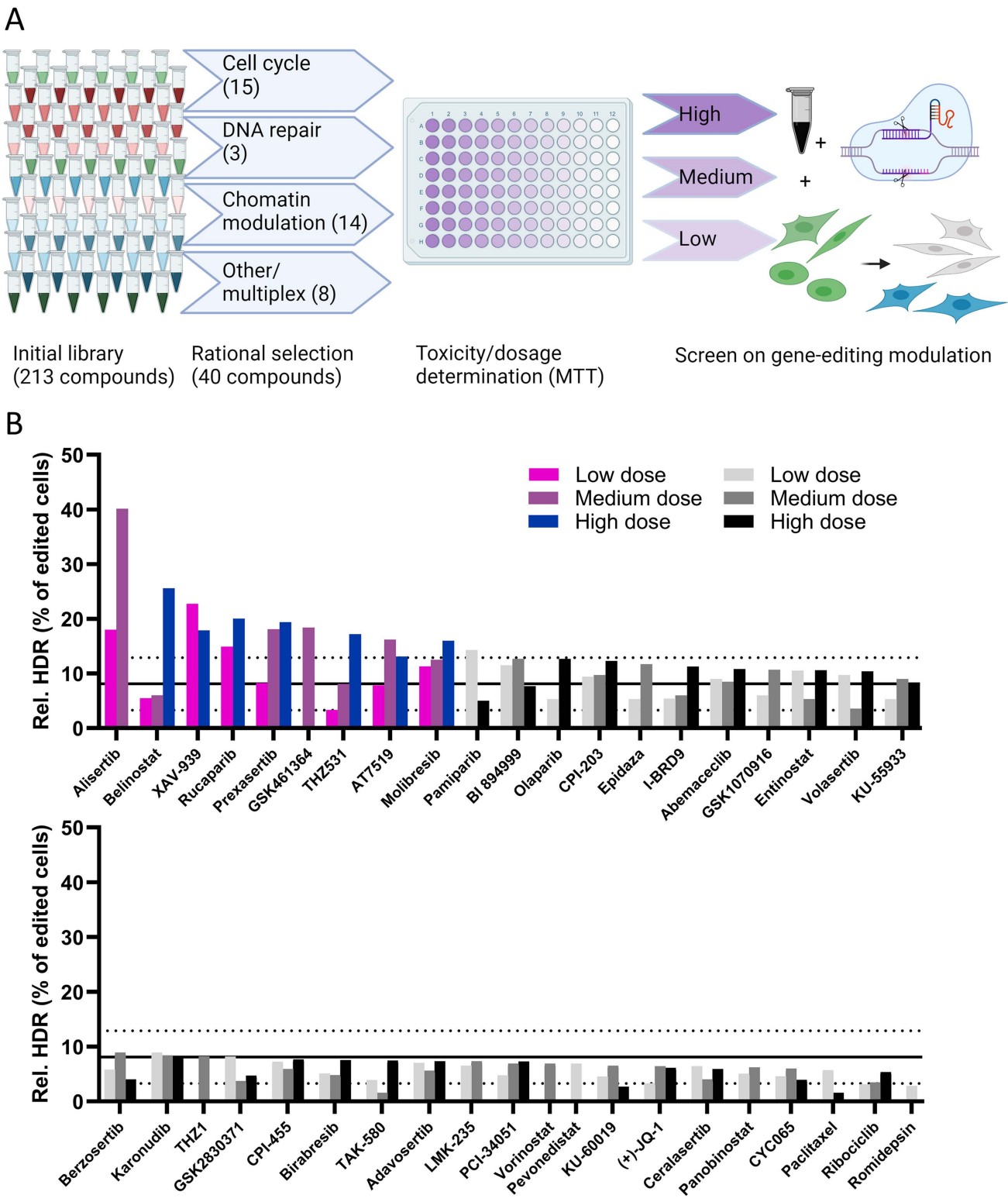

**Fig 1. Initial screening performed using oncological compounds on CRISPR genome editing outcomes.** A: Scheme showing the selection and screening process of the compounds for toxicity and gene-editing efficiency evaluation. Toxicity screening using a cell viability assay was done to find the dosages to be used in the subsequent drug screening for effects on gene-editing efficiency by conversion of eGFP positive cells to nonfluorescent

or blue fluorescent cells. B: Efficiency of HDR relative to all gene edited cells, in HEK293T-eGFP cells treated with compounds in up to three dosages based on toxicity screening: High ($^{10}$log value below IC50), Medium (10-fold lower than high dose) and Low (10-fold lower than medium dose). Efficacy was compared to cells treated with only CRISPR-Cas9 LNP (mean +- SD as solid and dotted lines; n = 29 technical replicates). Each bar represents one well of >1000 cells in the single-cell gate in flow cytometry. Colored bars were considered hits in this initial screening, and studied in further validation experiments (explained in text). Top: Best scoring half of the compounds. Bottom: Worst scoring half of the compounds.

and improved HDR, while belinostat increased both NHEJ and HDR, with a relatively pronounced increase for HDR in this study. Taken together, alisertib exhibited the strongest effect on both gene editing outcomes (NHEJ inhibition and HDR enhancement) compared to LNP-treated control cells. Between 0.1 and 0.3 µM the relative HDR incidence increased over 5-fold and became the preferred gene editing outcome (>50% relative HDR incidence) at 1 µM. Due to this drastic effect, a narrower dose-range was investigated, shown in S4 Fig in S1 File. The inhibitory effect on NHEJ was dose-dependent in this range, while HDR markedly increased between 0.2 and 0.3 µM. Thus 0.3 µM seemed to be the lowest effective concentration for preferential HDR activation in HEK293T-eGFP cells, while 1 µM caused HDR to become the most prominent repair pathway.

To assess the dose-dependency of alisertib on the CRISPR-Cas mediated gene editing outcome, cells were pretreated with either 0 or 1 µM alisertib 24 hours prior to LNP addition. LNPs were administered to cells with either 10 nM (standard dosage in other experiments) or 30 nM of SpCas9. When a higher dose of LNP was added, both gene knockout and gene correction populations increased proportionally to the dosage as shown in Fig 3A and Fig 3B. In the case of pre-treatment with 1 µM alisertib, the relative HDR incidence stayed above 50% indicating that with a higher total gene editing incidence, HDR was still the predominant pathway. Furthermore, when the alisertib incubation time was varied it showed that simultaneous treatment improved gene editing as well, with a 2.5-fold significant increase of relative HDR efficiency (S5 Fig B in S1 File). The toxicity of alisertib did not vary significantly when comparing different incubation times (S5 Fig A in S1 File).

HDR-mediated gene correction was further validated at the genetic level by amplifying the eGFP locus using PCR and subsequent sequencing of the amplicons. The sequencing traces were analyzed using the TIDER method (x31). This showed that at the genetic level, the relative HDR incidence was higher for alisertib primed cells as well. However, the total NHEJ and HDR incidences found by TIDER analysis were much higher than found in the fluorescent protein expression in flow cytometry (Fig 3C). The distribution of insertions and deletions revealed that most genotypes had a −3 deletion, which could explain this discrepancy as this may not lead to gene knockout in some cases (S6 Fig in S1 File).

An observation in these experiments was that cells treated with alisertib had a delayed cytotoxicity, which was not captured in the initial MTT assay. After 2 days, the cell viability as measured by MTS was not affected by alisertib. This is shown in Fig 3D as well as Fig 3E, in which the morphology is shown to resemble healthy HEK293T cells. However after 5 days, consistent with the duration of the experiments presented in this work, cells started exhibiting a dose-dependent decrease in cell viability. The confluency decreased, and cells with a disturbed morphology started appearing (Fig 3E, marked by the arrows). Seven days after treatment started, cells treated with at least 0.3 µM alisertib showed very low metabolic activity and confluency, and cell morphology was completely disrupted.

Hepa 1–6-eGFP cells, a murine hepatoma cell line, were used to further investigate whether the observed HDR preference was cell-line and species independent. Three Aurora kinase inhibitors were used with differing specificities for aurora kinases A, B and C, to assess the pathway specificity in parallel. Alisertib is selectively an AURKA inhibitor. PF-03814735 inhibits both AURKA and Aurora kinase B (AURKB) and danusertib is a pan-aurora kinase inhibitor of AURKA, AURKB and Aurora kinase C (AURKC). All three inhibited NHEJ up to 2-fold (Fig 4A) increased HDR up to 5 fold (Fig 4B). This resulted in a positive trend for improving relative HDR incidence similarly to HEK293T-eGFP cells, as shown in Fig 4C. Relative HDR increased 3-fold for alisertib in concentrations higher than 0.3 µM, and similar effects were seen for danusertib and PF-03814735. However, toxicity was a concern in these cells as well. The number of detected cells in flow

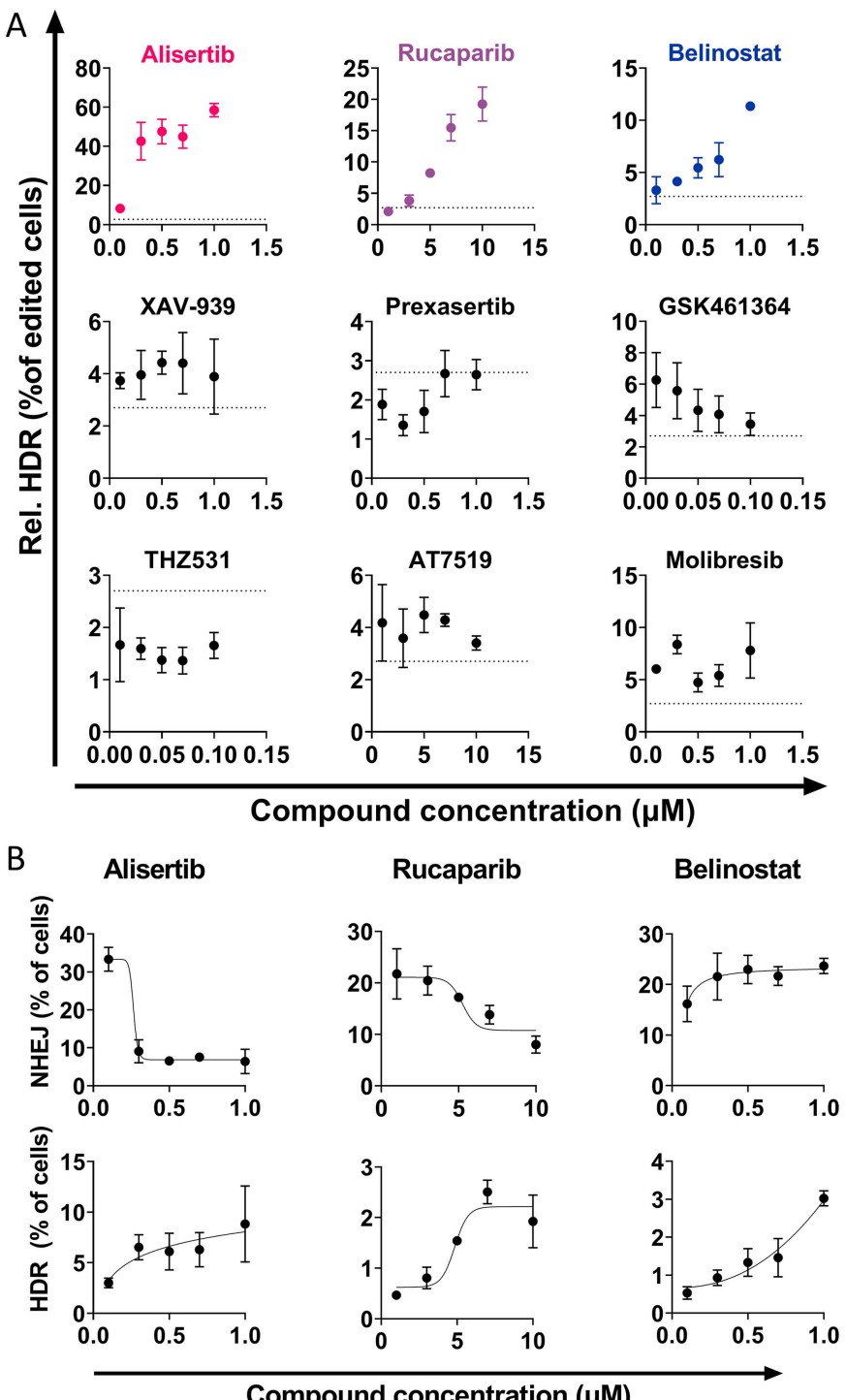

**Fig 2. Hit validation of the findings in Fig 1.** A: Repeated experiment in a narrower dose-range for the hits compared to LNP treatment without compounds (mean (dotted line); 2,9%). Of these, alisertib, rucaparib and belinostat yielded a clear dose-dependent HDR increase. B: The result of the three significant and dose-dependent hits from A separated into the gene knock-out (NHEJ, top) and correction (HDR, bottom) outcomes. All conditions represent n = 3 technical replicates.

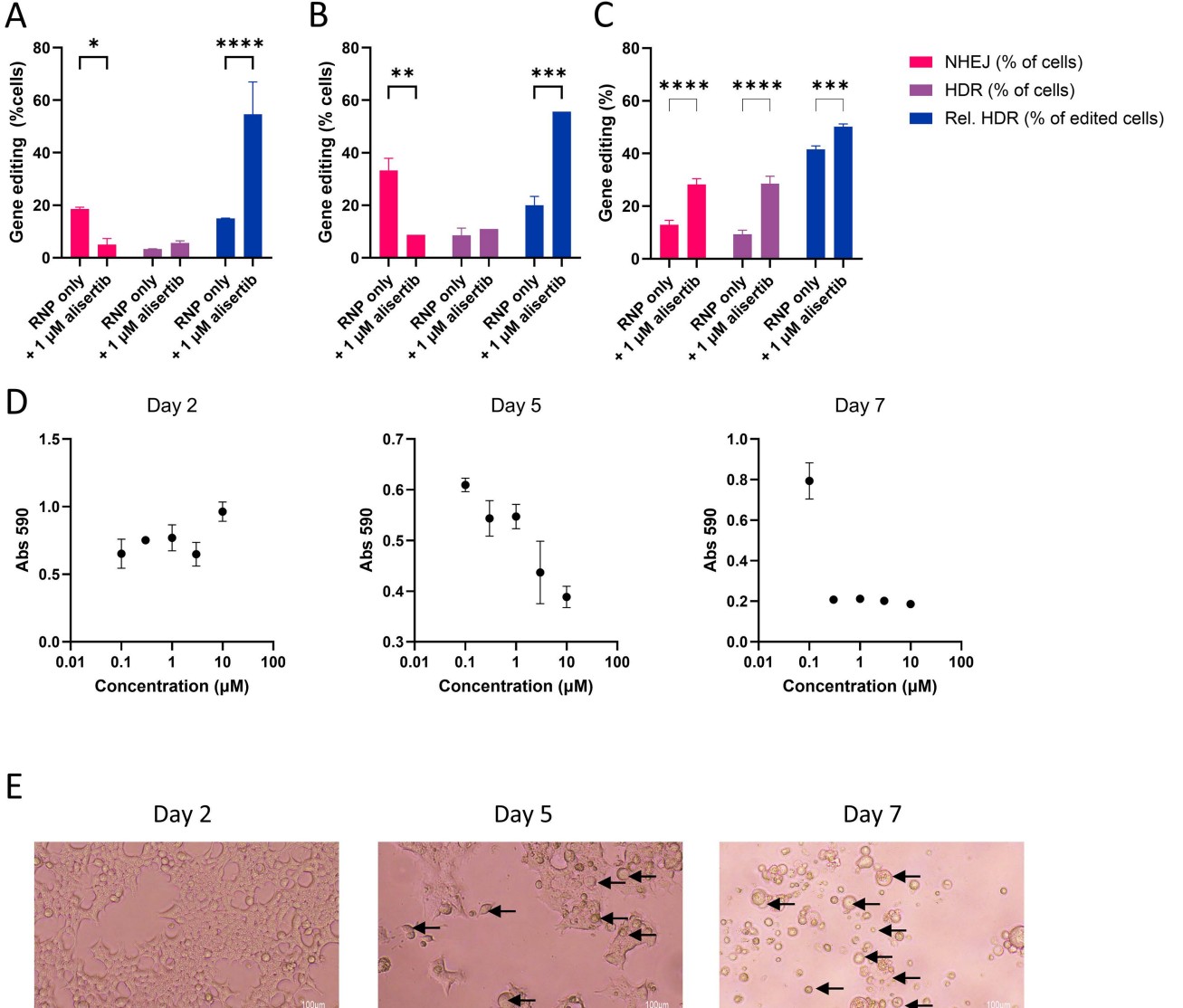

**Fig 3. Further validation of alisertib in HEK293T cells relating to CRISPR gene editing outcomes.** A and B: LNP dose-dependency in cells treated without or with 1 μM alisertib with 10 nM RNP (A) or 30 nM RNP **(B)**. C: TIDER gene-editing outcomes for cells treated with 10 nM CRISPR-Cas9 formulation alone or co-treated with 1 μM alisertib. C: Time-resolved toxicity of cells treated with alisertib at 0 days. Medium containing alisertib was replaced with standard culture medium on day 2. *: $p < 0.05$; **: $p < 0.01$; ***: $p < 0.001$; ****: $p < 0.0001$. All conditions represent n = 3 technical replicates.

cytometry decreased with higher dosages (Fig 4D), which was due to a reduced cell viability as measured by metabolic activity (Fig 4E). A dosage of 0.3 μM relatively showed overall high efficacy and manageable toxicity for alisertib and danusertib, while 0.2 μM was favorable for PF-03814735. Taken together alisertib had a strong effect (22.4% relative HDR incidence) for a relative cell viability of 37% at a concentration of 0.3 μM, which is the most favorable profile between the three inhibitors and the tested concentrations. Microscopy revealed that the cell morphology after treatment with 0.3 μM danusertib (Fig 4H) after 5 days was not disrupted compared to untreated control conditions (Fig 4F). The morphology using alisertib (Fig 4G) or PF-03814735 (Fig 4I) also did not change as drastically as it did for the HEK293T-eGFP cells, but the confluency of cells was noticeably lower than in the untreated control. The toxicity of alisertib on healthy primary

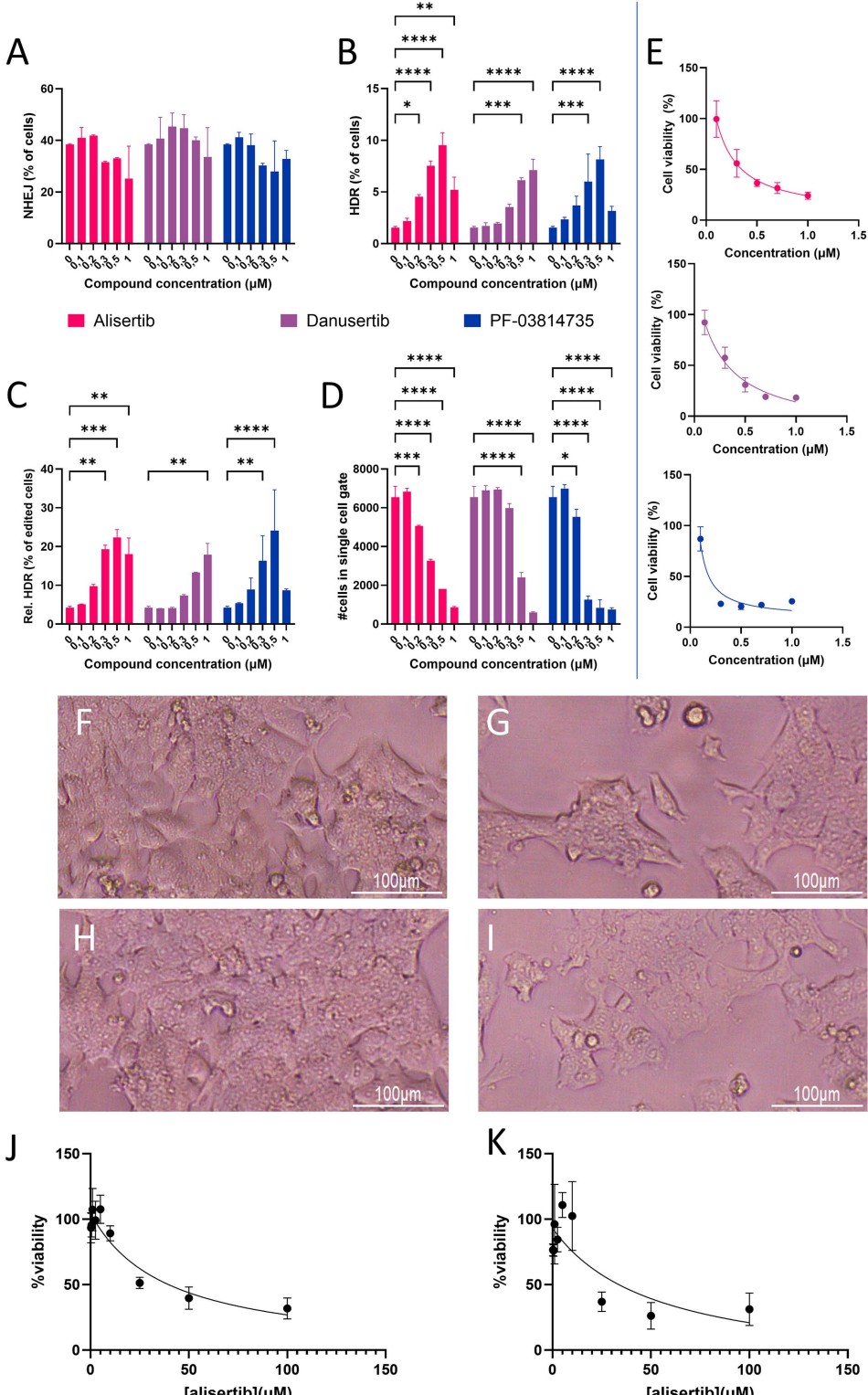

**Fig 4. Gene editing efficacies and cytotoxicity on Hepa 1-6 eGFP pretreated with aurora kinase inhibitors alisertib, danusertib or PF-03814735 using ProdeliverIN CRISPR for RNP delivery.** Colors are consistent between panels A-E. A: NHEJ incidence with ascending dosages of the three AURKA inhibitors. B: Absolute HDR incidence with ascending dosages of the thee AURKA inhibitors. C: Relative HDR incidence calculated from the

percentages in panels A and B, for cells treated with ascending dosages of the three AURKA inhibitors. D: Cell counts in the single cell gate in the flow cytometry data after acquiring 10.000 cells per condition, for cells treated with ascending dosages of the three AURKA inhibitors. *: $p < 0.05$; **: $p < 0.01$; ***: $p < 0.001$; ****: $p < 0.0001$. All conditions represent n = 3 technical replicates. E: Cell viability measured by MTS assay 5 days after the start of treatment. N = 6 technical replicates. F-I: Microscopic pictures of cells treated with no compound **(F)**, 0.3 µM of alisertib **(G)**, 0.3 µM danusertib (H) or 0.3 µM PF-03814735. J and K: Cell viability of an alisertib dose escalation on primary murine hepatocytes 48h (J) and 120h (k) after alisertib addition (total 48h incubation time), n = 6 technical replicates.

murine hepatocytes was finally determined as IC50 value at 48h and 120h after exposure. The IC50 values were 32.7 and 54.8 µM respectively (**Fig 4J and 4K**), indicating a similar toxicity profile at these timepoints.

Finally, the application of these findings on a disease-relevant locus, PCSK9, was investigated as detailed in **S1 Methods** in S1 File. The effect of alisertib on gene editing on the PCSK9 locus was investigated in Hepa 1–6 cells, sgRNA and template sequences are detailed in **S1 Table** in S1 File. The transfection was effective at targeting the eGFP gene, as observed in previous experiments (S7 Fig A in S1 File). On the PCSK9 locus, only modest gene editing was measured when the sequences were analyzed by TIDER. HDR was measurable only when alisertib was present, and the relative HDR incidence for this condition (15.8%) was in line with the previously observed HDR preference on the eGFP locus in Hepa 1–6 cells using 0.3 µM alisertib (19.3%).

## Discussion

The initial rationale of this screen was to find compounds that lead to HDR being favored over NHEJ, which can feasibly be given as a targeted, synergistic treatment with CRISPR-Cas9-based gene editing therapeutics. Oncological drugs were screened due to the similarities between the pathways targeted in cancer and those involved in genome editing, such as cell cycle regulation and DNA damage repair. The 40 selected compounds exhibit varied subcellular targets and processes as shown in Table 1. Many studies on small molecule CRISPR enhancers have been performed already, with varying success. For example, the DNA ligase 4 inhibitor SRC7 has been widely utilized [21,22]. However, this compound has not been used in any clinical trials, while many of the compounds investigated in this study are, or were, in various phases of clinical development.

The screen revealed many compounds that did not affect the outcome of gene editing significantly or relevantly, but also three that did show a favorable effect. Two out of three confirmed hits were reported to influence gene repair outcomes in previous studies. HDAC inhibitors, such as belinostat, have shown in the past to improve overall gene editing [32] and HDR specifically [33], due to their effects on chromatin packaging of the DNA. This efficacy was recently demonstrated for prime editing as well [34]. These compounds therefore served as an internal validation for the screen. Interestingly however, the other pan-HDAC inhibitors (entinostat, vorinostat and panobinostat) did not show the same effect. Other HDAC inhibitors were not reported previously. Epidaza, which inhibits HDAC 1–3, did not show an effect towards improving HDR and romidepsin, which inhibits HDAC 1 and 2, strongly inhibited HDR in this study compared to NHEJ. Further study on which HDAC subtypes inhibited by these compounds dictate genome editing outcomes is therefore needed.

Rucaparib, a PARP 1 and PARP 2 inhibitor involved in the DNA damage signaling checkpoint, affected the gene editing outcomes as well. Whereas rucaparib has previously been shown to improve gene editing due to inhibition of the microhomology-mediated end joining pathway [35], the observed effect on HDR found in the current study has not been reported to the best of our knowledge. Inhibition of PARP 1 and PARP 2 directly influences the regulation of base excision repair, which is usually a single stranded DNA damage event. However it is reported that inhibition of PARP-1 drives the cell towards homologous recombination, which in oncology is used to cause cell death in BRCA-deficient cells [36], and could explain our observations. Furthermore, this drug is approved for clinical use in humans, and therefore clinical knowledge exists to potentially devise a synergistic treatment plan for CRISPR-Cas and rucaparib combination therapy.

The primary discovery of the screening was the simultaneous NHEJ inhibition and HDR induction found when pretreating cells with alisertib. This compound is used in anti-cancer therapy to inhibit AURKA, which is involved in mitotic spindle formation and organization, and has been implicated in DNA signaling in cancers [37,38]. Reports on mechanisms in healthy cells are sparse, but the toxicity was shown to be lower in healthy cells than in breast cancer cells [39], which we noted in our study on primary human hepatocytes as well. The effects found on gene editing efficiency therefore needs to be investigated more on the mechanistic level to unravel this observed relationship between AURKA inhibition and HDR efficiency.

Addition of alisertib resulted in the greatest effect observed in this study, showing a preference for HDR over NHEJ outcomes in HEK293T-eGFP cells. This was seen on the phenotypic level by BFP expression compared to eGFP knock-out, and to a lesser extent on the genetic level shown by sequencing and TIDER analysis. This may be due to the predominant mutations found in TIDER being in-frame (−3), which may not disrupt the eGFP protein function. We found that treating cells with alisertib and treating them with a higher dosage of LNP increases the efficacy as well, validating further that the effect is due to priming the cells for CRISPR HDR by increasing the RNP and HDR template concentrations. If this pathway can be inhibited in a non-toxic way, it can therefore lead to greater specificity of gene correction.

In our initial screen we classified alisertib to be non-toxic, based on the IC50 gathered from the MTT assay data. However, when looking closer at the toxicity curve two days after treatment, a loss of 20% cell viability can be seen at a concentration of 0.1 μM (S2 Fig. in S1 File). This led us to scrutinize the toxicity in more detail. The toxicity becomes apparent 5 days after the start of alisertib treatment. This was independent of total compound incubation time, which was varied between 24 hours and 0 hours of pre-incubation of cells with alisertib (S5 Fig A in S1 File). The observed in the Hepa 1–6 cells after 5 days presented in Fig 3 are in line with the HEK293T results, which indicates that the toxicity was simultaneously species and cell type independent, at least in these model systems. Toxicity of these AURKA inhibitors needs to therefore be investigated further in more relevant cell types to assess if these effects are transient and significant. Toxicity in primary murine hepatocytes was in the order of 30–50 μM, and did not show a time-dependent worsening of the toxicity as we noted for HEK293T and Hepa 1–6 cells (Fig 4J–4K).

Finally, two other AURKA inhibitors (danusertib and PF-03814735) were assessed in Hepa 1–6-eGFP as well to validate the pathway. The manufacturer summarized the efficacy of these compounds towards AURKA, AURKB and AURKC. Of these, PF-03814735 is the most potent towards AURKA with an IC50 of 0.8 nM. Alisertib has potency in the same order of magnitude with an IC50 of 1.2 nM, and danusertib is magnitude less active at 13 nM. This is reflected in the efficacy, as danusertib requires a higher concentration before the effect on gene editing efficiency, as well as the toxicity, was visible, although the toxicity is in the same order of magnitude for all three compounds. Danusertib also has activity against AURKB and AURKC, with an IC50 of 79 and 61 respectively, whereas PF-03814735 has a preference for AURKB at an IC50 of 0.5 nM. GSK1070916, an AURKB and AURKC inhibitor, did not show an effect toward relative HDR activation, so these pathways likely only contribute to the cytotoxicity. PF-03814735 showed a clearly more drastic toxicity, likely due to the strong combined AURKA and AURKB inhibition. Danusertib and alisertib showed similar cytotoxicity, but the efficacy of alisertib was higher.

To assess the efficacy of alisertib further we designed sgRNA and a ssODN template for the murine PCSK9 gene, based on previously reported work [29]. At this stage we were unable to achieve robust genome editing in these cells on the PCSK9 locus using gene sequencing and TIDER, while editing the eGFP locus in Hepa 1–6 cells using the same transfection mix did show 40% gene knock-out (S7 Fig in S1 File). This aligns with reported limits of the detection limit in Sanger-sequencing based gene editing analysis methods [40]. Only the samples treated using 0.3 μM alisertib exhibited measurable gene editing above the noise, with the relative HDR incidence being comparable to the findings on the eGFP locus in Fig 4C. An explanation of the lower overall editing on the PCSK9 locus could be that this endogenous locus is epigenetically more associated in nucleosomes compared to the eGFP transgene, which is known to inhibit Cas9 genome

editing [41]. Further optimization is therefore needed on disease relevant targets and cell types before the added benefit of alisertib can be assessed, which is outside the scope of the current work. Furthermore, validation is needed to assess the effects of gene editing in presence of alisertib on off-target sites and, especially, on the whole genome, as other potent compounds have shown major genomic alterations occurring upon treatment with DNA-PK inhibitors [42].

## Conclusion

Of the forty screened compounds, three showed a significant HDR enhancing effect: belinostat, rucaparib and alisertib. Alisertib specifically shows a rapid onset of action to this end, as well as activity in a relevant cell line. While AURKA inhibition showed a relevant increase of HDR, the toxicity displayed in this study limits its application. Other means of AURKA inhibition might be effective and warrants further investigation. Furthermore, targets downstream of AURKA should be further investigated to find the specific drivers of this effect, and allow application in an HDR-based gene editing approach in a more relevant setting such as primary cells or *in vivo*, which should be applied on a broader range of disease-relevant target genes.

## Supporting information

**S1 File. Additional details on our analysis procedures, alternative analysis of the main screen data and supporting data which led to the figures presented in the main text.**
(DOCX)

## Acknowledgments

The authors would like to acknowledge Omnia Elsharkasy for her help with cell sorting to select high-expressing Hepa 1–6 eGFP cells. Furthermore the authors acknowledge Dr. Yanjuan Xu for their help in preparing the primary murine hepatocytes.

## Author contributions

**Conceptualization:** Danny Wilbie, Selma Eising, Jan J Molenaar, Enrico Mastrobattista.

**Formal analysis:** Selma Eising.

**Investigation:** Danny Wilbie, Vicky Amo-Addae, Esmeralda Bosman.

**Methodology:** Danny Wilbie, Selma Eising, Vicky Amo-Addae, Johanna Walther, Esmeralda Bosman, Zhiyong Lei.

**Resources:** Zhiyong Lei, Olivier G de Jong, Jan J Molenaar.

**Supervision:** Jan J Molenaar, Enrico Mastrobattista.

**Writing – original draft:** Danny Wilbie.

**Writing – review & editing:** Olivier G de Jong, Enrico Mastrobattista.

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
