## [Decision Letter · Decision Letter 0]

20 Jan 2025

PONE-D-24-57469Anti-cancer compound screening identifies Aurora Kinase A inhibition as a means to favor CRISPR/Cas9 gene correction over knock-outPLOS ONE

Dear Dr. Mastrobattista,

Thank you for submitting your manuscript to PLOS ONE. After careful consideration, we feel that it has merit but does not fully meet PLOS ONE’s publication criteria as it currently stands. Therefore, we invite you to submit a revised version of the manuscript that addresses the points raised during the review process.

We look forward to receiving your revised manuscript.

Kind regards,

Amir Faisal, PhD

Academic Editor

PLOS ONE

Journal requirements: When submitting your revision, we need you to address these additional requirements. 1. Please ensure that your manuscript meets PLOS ONE's style requirements, including those for file naming. The PLOS ONE style templates can be found at https://journals.plos.org/plosone/s/file?id=wjVg/PLOSOne_formatting_sample_main_body.pdf and https://journals.plos.org/plosone/s/file?id=ba62/PLOSOne_formatting_sample_title_authors_affiliations.pdf. 2. Please include a caption for figure 5 and 6.  3. PLOS requires an ORCID iD for the corresponding author in Editorial Manager on papers submitted after December 6th, 2016. Please ensure that you have an ORCID iD and that it is validated in Editorial Manager. To do this, go to ‘Update my Information’ (in the upper left-hand corner of the main menu), and click on the Fetch/Validate link next to the ORCID field. This will take you to the ORCID site and allow you to create a new iD or authenticate a pre-existing iD in Editorial Manager. 4. Please include captions for your Supporting Information files at the end of your manuscript, and update any in-text citations to match accordingly. Please see our Supporting Information guidelines for more information: http://journals.plos.org/plosone/s/supporting-information.

Reviewers' comments:

Reviewer's Responses to Questions

**Comments to the Author**

1. Is the manuscript technically sound, and do the data support the conclusions?

Reviewer #1: Yes

Reviewer #2: Partly

Reviewer #3: Partly

2. Has the statistical analysis been performed appropriately and rigorously? 

Reviewer #1: Yes

Reviewer #2: Yes

Reviewer #3: I Don't Know

3. Have the authors made all data underlying the findings in their manuscript fully available?

Reviewer #1: Yes

Reviewer #2: Yes

Reviewer #3: No

4. Is the manuscript presented in an intelligible fashion and written in standard English?

Reviewer #1: Yes

Reviewer #2: Yes

Reviewer #3: Yes

5. Review Comments to the Author

Reviewer #1: The authors present here a mini screen of 40 oncological drug compounds with the aim to enhance the efficiency of the homology-directed repair (HDR) pathway and identified three—compounds, rucaparib, belinostat, and alisertib, effectively showing improved HDR. Notably, alisertib demonstrated a more than fourfold increase in precise gene correction in specific cell models, however, it also showed some cytotoxicity, indicating a need for further investigation into its mechanisms and effects. While the results are nicely presented, some further clarification and some data are required in the form of a revised manuscript before it can be formally accepted for publication. Below are my specific comments.

1. It is not clear if the viral transduced eGFP cells were single-cell sorted or a population. Ideally, a single-cell clone should have been chosen and checked for the number of GFP transgene copies through qPCR.

2. Also, this analysis should have been conducted on an endogenous gene, perhaps endogenously tagged with GFP. For this analysis, HEK293T cells would not be an ideal model system because of well-known ‘ploidy’ issues. MCF10A would have been a better choice for a cellular model system as they are considered to be near diploid and also have efficient genome maintenance dynamics.

3. Furthermore, the study would have been more clinically relevant if authors considered an actual disease model system, and targeted mutated genes in diseases such as Fanconi anemia (FA), Ataxia Telangiectasia (AT), etc., and corrected the mutation in the presence or absence of this drug, as a final figure. This could be a good way of ex-vivo CRISPR-Cas9-based gene editing for a variety of human diseases.

4. The toxicity of Alisertib can be reduced by omitting the pre-incubation of the drug. Figure S5 shows a very small difference in the relative HDR incidence in the conditions of no-preincubation and incubation for 6-24h with Alisertib. Also, the authors should test if the delayed toxicity is reduced by omitting the pre-incubation step through MTT or similar assay (toxicity data is not shown in Figure S5).

5. Figure 2A indicates that Alisertib can be used at even a lower nanomolar dose than the lowest one used in the study. I am wondering if the authors go from the lowest dose used (i.e., 100 nM) further down to 1-10 nM conc. of Alisertib and check the relative HDR efficiency. This may further improve the cellular toxicity observed in the tested model cellular systems, while still keeping a reasonable HRD efficiency.

6. While some studies (PMID: 37580318; PMID: 37537500) show a positive outcome of using DNA-PK inhibitor towards HDR editing efficiency, a more recently published article (PMID: 39604565) shows that using a potent DNA-PK inhibitor causes frequent large-scale genomic alterations including deletions and translocations. It is, therefore, important that any study publishing a positive (or negative impact) of a compound/drug on CRISPR-Cas9-based genome editing must consider a whole genome analysis, preferably on single cells to gauge the adverse genomic events caused by the drug, in addition to being effective for HDR. The authors should address this in the discussion section.

7. Figure 3B is not referenced in the text.

Reviewer #2: In the manuscript titled "Anti-cancer compound screening identifies Aurora Kinase A inhibition as a means to favor CRISPR/Cas9 gene correction over knock-out", authors have described results from a screening of rationally selected 40 anti-cancer compounds in HEK29T cell line to identify ones that result in increased Cas9-mediated HDR correction. Authors found that Aurora Kinase A inhibitor alisertib increases HDR efficiency by up to 4-folds but results in significant cytotoxicity. Three Aurora Kinase inhibitors were further validated in Hepa1-6 line for determining broader applicability of the compounds. The findings from this study would undoubtedly be useful for clinical applications of cell and gene therapies; not only for in vivo therapeutics that focus on gene correction (a use case that authors identify in the main text) but also for ex vivo therapies - such as both autologous and allogenic CAR-T cell therapies that could benefit from precise integration of the CAR at a safe harbor genomic loci in the target cells. However, the results presented in their current form do not fully align with the objectives originally set forth by the authors in this manuscript.

Major considerations:

1. Authors have to demonstrate the suitability of these compounds for clinically relevant cell types. Authors should conduct testing of these compounds in T-cells or pluripotent stem cells to assess the risk-benefit profile. In a clinically relevant cell line, if authors are able to demonstrate significant increase in HDR efficiency without significant impact on cellular health - then these compounds have meaningful relevance for clinical applications. Otherwise, these compounds might result in increased cytotoxicity of the intended cellular population and any early apparent increment in HDR may not result in actual clinical benefit forgoing the premise of this manuscript.

2. Authors concluded that eGFP to BFP conversion reporter system may not be an ideal choice for their intended setup as it overestimates the relative HDR efficiency. In Fig 3A vs Fig 3B, there is a clear discrepancy on relative HDR% as estimated by flow cytometry and TIDER analysis respectively with flow overestimating the relative %HDR and TIDER underestimating it. The ground truth needs to be determined by testing HDR efficiencies at 2-3 different genomic sites such as safe harbor sites (AAVS1, CCR5 etc) and donor and gRNA combinations.

If authors can address the above mentioned points, I am happy to take another look at the revised manuscript however I do recognize that this is a major effort. I firmly believe that addressing the aforementioned issues will lead to a meaningful contribution for the field of cell and gene therapy and the readers of PLOS One.

Reviewer #3: Anti-cancer compound screening identifies Aurora Kinase A inhibition as a means to favor CRISPR/Cas9 gene correction over knock-out

This paper explores the challenges and advancements in enhancing HDR for gene therapy using CRISPR technology, focusing on the potential of oncological drugs to shift the repair pathway preference from NHEJ to HDR. A screening of 40 compounds identified three key drugs—alisertib, rucaparib, and belinostat—that significantly improved HDR efficiency in two cell lines, with alisertib showing the most promise despite its associated cytotoxicity. The findings suggest that these compounds could enhance CRISPR-mediated gene editing outcomes, although further investigation into their mechanisms and long-term effects is necessary due to the observed delayed cytotoxicity.

The repurposing of approved drugs is always a positive, the work is exciting and largely well presented. I think this is a good start and the following are my comments and suggestions.

1. From either the text or the figures, it is difficult to decipher if biological and/or technical replicates were performed for all the experiments. Please clarify.

Statistics were rarely performed and there is no indication of what the stats were, when performed; this further leads me to ask the question about appropriate replication with controls.

May be all of this was done, it is just not clear from the figures or the text, so please clarify.

2. Pre-defined abbreviations and consistency help the reader:

Example: Page 1: “This ribonucleoprotein complex binds to the DNA sequence complementary to the guide RNA, after which the Cas9 nuclease causes a double stranded DNA break (DSB). In the context of gene editing in cells, the RNP needs to reach the cell nucleus and bind to its target”

RNP could have been defined within parentheses at the beginning of the sentence.

Another such instance is for DSB that can be pre-defined and used later on in the text.

Consistency issues in using CRISPR/Cas9 vs CRISPR-Cas9 although CRISPR/Cas9 is more appropriate.

3. Page 2: Please add reference(s) for

“. While autologous gene-corrected cells have recently entered clinical trials, this drawback has led the field to consider alternative gene-editing tools for direct in vivo injection of HDR machineries.”

4. Page 3: A typo perhaps under the heading “Hepa 1-6 eGFP cell line construction”

The cell line should be Hepa 1-6 unlike as stated here “ at a 2:1:1 ratio in HEK293T cells using”

5. Page 4: Under the heading “Cytotoxicity assays”

“Forty microliters of a HEK293t-EGFP” where the T should be capitalized.

6. I think this is really cool :

“Briefly, a ssDNA template was used carrying two nucleotide mutations to convert the eGFP sequence to that of a blue fluorescent protein (BFP), ”

7. Pet peeve, perhaps??

Different scales on the Y-axis in the same figure makes it harder to digest and interpret the data. Might want to reconsider!

8. Page 12: Under “Gene sequencing and genotype analysis

For genotypic analysis, HEK293T-eGFP cells were treated with alisertib for 72 hours and CRISPR LNP for 48 hours prior to harvesting by trypsinization.”

It is unclear if the treatments were done subsequently or concurrently or something else? Please clarify briefly….

9. Figure 3B is not mentioned in the text, should be on Page 12 in the penultimate paragraph

“However, the total NHEJ and HDR incidences found by TIDER analysis were much higher than found in the fluorescent protein expression in flow cytometry (Figure 3B?). The”

10. At the beginning of the conclusion:

“Other means of AURKA inhibition might be effective and warrants further investigation.”

It would help to mention what other means very briefly as it would help the uninitiated reader….

11. Figure 1B is very dense and not all conditions have data, perhaps faceting the data either by dose or by drug might lend clarity to deciphering what bars belong to which drug! Please consider…

12. I am not sure if all the raw data were also submitted in spreadsheet format or other formats as data transparency is now an expected thing. Please consider submitting, if not done already.

13. There are recent papers that have shown that using drugs to enhance HDR causes genome instability in unexpected ways, the authors of this paper should discuss these issues and include the relevant references as well.

Example: https://www.nature.com/articles/s41587-024-02488-6

6. PLOS authors have the option to publish the peer review history of their article (what does this mean? ). If published, this will include your full peer review and any attached files.

**Do you want your identity to be public for this peer review?** For information about this choice, including consent withdrawal, please see our Privacy Policy .

Reviewer #1: No

Reviewer #2: No

Reviewer #3: No

---

## [Author Response · Author response to Decision Letter 1]

20 Jul 2025

Reviewer #1

The authors present here a mini screen of 40 oncological drug compounds with the aim to enhance the efficiency of the homology-directed repair (HDR) pathway and identified three—compounds, rucaparib, belinostat, and alisertib, effectively showing improved HDR. Notably, alisertib demonstrated a more than fourfold increase in precise gene correction in specific cell models, however, it also showed some cytotoxicity, indicating a need for further investigation into its mechanisms and effects. While the results are nicely presented, some further clarification and some data are required in the form of a revised manuscript before it can be formally accepted for publication. Below are my specific comments.

1. It is not clear if the viral transduced eGFP cells were single-cell sorted or a population. Ideally, a single-cell clone should have been chosen and checked for the number of GFP transgene copies through qPCR.

This is indeed an important point, as the number of genomic eGFP integrations may affect the analysis of this read-out. We have chosen to infect both cell lines at a low multiplicity of infection (MOI = 0.1) and infer from this that there is likely only one copy of the eGFP gene in the cells. For the Hepa 1-6-eGFP cells we have described this on page 6 line 132-133.

2. Also, this analysis should have been conducted on an endogenous gene, perhaps endogenously tagged with GFP. For this analysis, HEK293T cells would not be an ideal model system because of well-known ‘ploidy’ issues. MCF10A would have been a better choice for a cellular model system as they are considered to be near diploid and also have efficient genome maintenance dynamics.

To explain our initial rationale, we decided to first use a cell line which can be readily transfected with CRISPR-Cas9 at high and reproducible rates, hence the use of HEK293T for the initial screen. We subsequently chose Hepa 1-6-eGFP to move toward a primary liver cell line, as liver tissue presents a relevant and viable target for future clinical work. To assess effects over multiple cell types without interference of potential ploidy issues, we chose to make use of eGFP cell lines transduced with a low multiplicity of infection (MOI = 0.1). As such, the eGFP gene is unlikely to be integrated more than once per cell.

That being said, we agree that including an additional endogenous target would increase the impact of this manuscript. To address the point of endogenous genes, we chose to edit PCSK9 in the Hepa 1-6 cells based used previously based on previous work validating the sgRNA and locus as shown in S7 Fig. While the results were an order of magnitude lower than for the eGFP locus, we demonstrate that HDR is only detected when alisertib is present, demonstrating an effect on an endogenous locus as well.

3. Furthermore, the study would have been more clinically relevant if authors considered an actual disease model system, and targeted mutated genes in diseases such as Fanconi anemia (FA), Ataxia Telangiectasia (AT), etc., and corrected the mutation in the presence or absence of this drug, as a final figure. This could be a good way of ex-vivo CRISPR-Cas9-based gene editing for a variety of human diseases.

We agree that the prospect of utilizing these results for ex-vivo cell engineering and curing disease is exciting. However, as specific disease models may have additional, potentially unexpected, effects on the outcome of gene editing assays, we feel that inclusion of the suggested disease models fall outside of the scope of this manuscript. However, as previously mentioned we do agree that assessing the effects on an endogenous gene increase the relevance and impact of this work. To address this we added the aforementioned S7 Fig on the PCSK9 gene as a proof of concept for future in vivo editing on a disease relevant gene locus. We furthermore added a point about disease models in the Discussion section: “At this stage we were unable to achieve robust genome editing in these cells on the PCSK9 locus in the Hepa 1-6 cell line using gene sequencing and TIDER, while editing the eGFP locus in Hepa 1-6 cells using the same transfection mix did show 40% gene knock-out (S7 Fig.). This suggests that further optimization is needed on disease relevant targets and cell types before the added benefit of alisertib can be assessed, which is outside the scope of the current work”.

4. The toxicity of Alisertib can be reduced by omitting the pre-incubation of the drug. Figure S5 shows a very small difference in the relative HDR incidence in the conditions of no-preincubation and incubation for 6-24h with Alisertib. Also, the authors should test if the delayed toxicity is reduced by omitting the pre-incubation step through MTT or similar assay (toxicity data is not shown in Figure S5).

We thank you for pointing out this lack of toxicity data. We require at least 24h of incubation for robust RNP transfection, meaning at least a 24h incubation with alisertib in the current experimental setup. Based on your suggestion we have added new data demonstrating that there is no significant toxicity difference for total alisertib incubation times between 24-72h in S5 Fig A.

5. Figure 2A indicates that Alisertib can be used at even a lower nanomolar dose than the lowest one used in the study. I am wondering if the authors go from the lowest dose used (i.e., 100 nM) further down to 1-10 nM conc. of Alisertib and check the relative HDR efficiency. This may further improve the cellular toxicity observed in the tested model cellular systems, while still keeping a reasonable HRD efficiency.

We did not investigate a lower dosage range as the effect of alisertib on especially the relative incidence of HDR only started between 0.1-0.15 µM (S4 Fig), and as such we do not expect an effect if we dose lower. We deem the effects seen in these lower dosages to be too low to be relevant for a real life application, and therefore did not investigate lower dosages here.

6. While some studies (PMID: 37580318; PMID: 37537500) show a positive outcome of using DNA-PK inhibitor towards HDR editing efficiency, a more recently published article (PMID: 39604565) shows that using a potent DNA-PK inhibitor causes frequent large-scale genomic alterations including deletions and translocations. It is, therefore, important that any study publishing a positive (or negative impact) of a compound/drug on CRISPR-Cas9-based genome editing must consider a whole genome analysis, preferably on single cells to gauge the adverse genomic events caused by the drug, in addition to being effective for HDR. The authors should address this in the discussion section.

We thank you for pointing out this recent literature. We have now addressed this point in the Discussion section on Page 25 line 492-495:

“Furthermore, validation is needed to assess the effects of gene editing in presence of alisertib on off-target sites and, especially, on the whole genome, as other potent compounds have shown major genomic alterations occurring upon treatment with DNA-PK inhibitors (42).”.

7. Figure 3B is not referenced in the text.

We thank you for noticing this, the error has been corrected on Page 18, Line 364.

Reviewer #2

Major considerations:

1. Authors have to demonstrate the suitability of these compounds for clinically relevant cell types. Authors should conduct testing of these compounds in T-cells or pluripotent stem cells to assess the risk-benefit profile. In a clinically relevant cell line, if authors are able to demonstrate significant increase in HDR efficiency without significant impact on cellular health - then these compounds have meaningful relevance for clinical applications. Otherwise, these compounds might result in increased cytotoxicity of the intended cellular population and any early apparent increment in HDR may not result in actual clinical benefit forgoing the premise of this manuscript.

We thank you for your constructive and valuable suggestion. We assessed the toxicity of our compound in primary murine hepatocytes in Fig 4, panels J and K and noted a more favorable toxicity profile than we see in HEK293T and Hepa 1-6 cells, especially when considering that there was a lack of delayed toxicity seen in the similar viability profiles over time. Further investigation of the gene editing in these cells is therefore an interesting follow up study, but due to the complexity of Cas9 RNP delivery in primary cells this is outside the scope of this current study. We chose Hepa 1-6, a primary liver tumor cell line, especially to prepare for this further application.

2. Authors concluded that eGFP to BFP conversion reporter system may not be an ideal choice for their intended setup as it overestimates the relative HDR efficiency. In Fig 3A vs Fig 3B, there is a clear discrepancy on relative HDR% as estimated by flow cytometry and TIDER analysis respectively with flow overestimating the relative %HDR and TIDER underestimating it. The ground truth needs to be determined by testing HDR efficiencies at 2-3 different genomic sites such as safe harbor sites (AAVS1, CCR5 etc) and donor and gRNA combinations.

We agree that no measurement method for gene editing is perfect and as such more validation is required. We have additionally investigated gene editing on the PCSK9 locus in Hepa 1-6 cells (S7 Fig.). We note that our positive control for genome editing (eGFP sgRNA and template DNA, measured by flow cytometry) shows robust knock-out here, while our design for PCSK9 showed a lower overall gene editing effect, potentially due to the epigenetic availability of this endogenous gene compared to transgenes introduced using lentiviral vectors. Unfortunately, sequencing-based methods such as TIDE and TIDER exhibit high limits of detection before gene editing can be shown (as shown in DOI: 10.3390/mps8020023) . We have added discussion on this to the discussion section:

“At this stage we were unable to achieve robust genome editing in these cells on the PCSK9 locus using gene sequencing and TIDER, while editing the eGFP locus in Hepa 1-6 cells using the same transfection mix did show 40% gene knock-out (S7 Fig). This aligns with reported limits of the detection limit in Sanger-sequencing based gene editing analysis methods (40). Only the samples treated using 0.3 µM alisertib exhibited measurable gene editing above the noise, with the relative HDR incidence being comparable to the findings on the eGFP locus in Fig 4C. An explanation of the lower overall editing on the PCSK9 locus could be that this endogenous locus is epigenetically more associated in nucleosomes compared to the eGFP transgene, which is known to inhibit Cas9 genome editing (41). Further optimization is therefore needed on disease relevant targets and cell types before the added benefit of alisertib can be assessed, which is outside the scope of the current work.”

However, we measured HDR efficiency and overall gene editing above the background measured in the controls only in cells treated with alisertib, with relative HDR occurring in line with previous our finding on the eGFP locus (15% relative HDR activation).

Further optimization of the gene editing conditions in a relevant cell type would therefore be interesting but is outside the scope of this work, which is to present the conducted screen with transparent limitations for data interpretation.

If authors can address the above mentioned points, I am happy to take another look at the revised manuscript however I do recognize that this is a major effort. I firmly believe that addressing the aforementioned issues will lead to a meaningful contribution for the field of cell and gene therapy and the readers of PLOS One.

We thank the reviewer for their positive words, and hope to have satisfied your concerns.

Reviewer #3

1. From either the text or the figures, it is difficult to decipher if biological and/or technical replicates were performed for all the experiments. Please clarify.

Statistics were rarely performed and there is no indication of what the stats were, when performed; this further leads me to ask the question about appropriate replication with controls.

May be all of this was done, it is just not clear from the figures or the text, so please clarify.

We thank you for this critical and constructive comment. Most of the figures here report n=3 technical replicates, which is now specified accordingly. We do however note the same effect in subsequent experiments, indicating that the effect of alisertib on both cell lines is robust and reproducible.

Regarding statistical analysis, we have added a section on our analyses in the Materials and Methods section:

“Statistical analysis was conducted using Graphpad PRISM version 9.3.1 using the Two-way ANOVA method, comparing the intra-group effect where relevant in a multiple comparisons test with Šidák correction unless stated otherwise. IC50 values were calculated using the [antagonist] vs response function of Graphpad PRISM version 9.3.1 where applicable”.

We have added analysis in relevant datasets and added statistical analysis details in figures 3A and 4 A-D, as wel as in the supplementary information.

2. Pre-defined abbreviations and consistency help the reader:

Example: Page 1: “This ribonucleoprotein complex binds to the DNA sequence complementary to the guide RNA, after which the Cas9 nuclease causes a double stranded DNA break (DSB). In the context of gene editing in cells, the RNP needs to reach the cell nucleus and bind to its target”

RNP could have been defined within parentheses at the beginning of the sentence.

Another such instance is for DSB that can be pre-defined and used later on in the text.

Consistency issues in using CRISPR/Cas9 vs CRISPR-Cas9 although CRISPR/Cas9 is more appropriate.

We thank you for these critical notes, and have improved the consistency in using abbreviations throughout the manuscript.

3. Page 2: Please add reference(s) for

“. While autologous gene-corrected cells have recently entered clinical trials, this drawback has led the field to consider alternative gene-editing tools for direct in vivo injection of HDR machineries.”

Thank you for pointing out this lacking reference, we have added a recent review paper and the recent report on single-patient base editing (DOI: 10.3389/fgeed.2025.1553590 and 10.1056/NEJMoa2504747 respectively) to back up this claim. We have also corrected this line to read “consider alternative gene-editing tools for direct in vivo injection of gene correction machineries”.

4. Page 3: A typo perhaps under the heading “Hepa 1-6 eGFP cell line construction”

The cell line should be Hepa 1-6 unlike as stated here “ at a 2:1:1 ratio in HEK293T cells using”

This line refers to construction of the lentivirus that was used to later construct the Hepa 1-6-eGFP cells. The virus itself was produced in HEK293T cells and subsequently purified and used on Hepa 1-6 cells.

5. Page 4: Under the heading “Cytotoxicity assays”

“Forty microliters of a HEK293t-EGFP” where the T should be capitalized.

We thank you for pointing out this mistake, we have corrected it accordingly.

6. I think this is really cool :

“Briefly, a ssDNA template was used carrying two nucleotide mutations to convert the eGFP sequence to that of a blue fluorescent protein (BFP), ”

We respectfully agree with this notion.

7. Pet peeve, perhaps??

Different scales on the Y-axis in the same figure makes it harder to digest and interpret the data. Might want to reconsider!

We agree and have normalized y-axes where we deemed this relevant and appropriate.

8. Page 12: Under “Gene sequencing and genotype analysis

For genotypic analysis, HEK293T-eGFP cells were treated with alisertib for 72 hours and CRISPR LNP for 48 hours prior to harvesting by trypsinization.”

It is unclear if the treatments were done subsequently or concurrently or something else? Please clarify briefly….

We thank you for pointing out the ambiguity in this phrasing. The treatment was done concurrently, meaning a 24h pre-incubation followed by transfection, thus yielding a 48h concurrent incubation of alisertib and LNP. This has been corrected in the quoted line.

9. Figure 3B is not mentioned in the

---

## [Decision Letter · Decision Letter 1]

3 Sep 2025

Anti-cancer compound screening identifies Aurora Kinase A inhibition as a means to favor CRISPR/Cas9 gene correction over knock-out

PONE-D-24-57469R1

Dear Dr. Mastrobattista,

We’re pleased to inform you that your manuscript has been judged scientifically suitable for publication and will be formally accepted for publication once it meets all outstanding technical requirements.

Kind regards,

Amir Faisal, PhD

Academic Editor

PLOS ONE

**Comments to the Author**

1. If the authors have adequately addressed your comments raised in a previous round of review and you feel that this manuscript is now acceptable for publication, you may indicate that here to bypass the “Comments to the Author” section, enter your conflict of interest statement in the “Confidential to Editor” section, and submit your "Accept" recommendation.

Reviewer #1: All comments have been addressed

Reviewer #3: All comments have been addressed

2. Is the manuscript technically sound, and do the data support the conclusions?

Reviewer #1: Yes

Reviewer #3: Yes

3. Has the statistical analysis been performed appropriately and rigorously? 

Reviewer #1: Yes

Reviewer #3: Yes

4. Have the authors made all data underlying the findings in their manuscript fully available?

Reviewer #1: Yes

Reviewer #3: Yes

5. Is the manuscript presented in an intelligible fashion and written in standard English?

Reviewer #1: Yes

Reviewer #3: Yes

6. Review Comments to the Author

Reviewer #1: The authors have adequately addressed all of my concerns. I recommend publishing this manuscript in PLOS One journal.

Reviewer #3: (No Response)

7. PLOS authors have the option to publish the peer review history of their article (what does this mean? ). If published, this will include your full peer review and any attached files.

**Do you want your identity to be public for this peer review?** For information about this choice, including consent withdrawal, please see our Privacy Policy .

Reviewer #1: No

Reviewer #3: No

---

## [Editor Report · Acceptance letter]

PONE-D-24-57469R1

PLOS ONE

Dear Dr. Mastrobattista,

I'm pleased to inform you that your manuscript has been deemed suitable for publication in PLOS ONE. Congratulations! Your manuscript is now being handed over to our production team.

Kind regards,

on behalf of

Dr. Amir Faisal

Academic Editor

PLOS ONE